# A Development of the Self Shape Adjustment Cushion Mechanism for Improving Sitting Comfort

**DOI:** 10.3390/s21237959

**Published:** 2021-11-29

**Authors:** Sooho Choi, Hyomin Kim, Hyungjoo Kim, Woosung Yang

**Affiliations:** 1School of Robotics, Kwangwoon University, 60 Gwangun-ro, Nowon-gu, Seoul 01890, Korea; crazineer@gmail.com (S.C.); hyomin.bicar@gmail.com (H.K.); 2Vitesco Technologies Korea, 45-29 Saeum-Ro, Icheon-si 17308, Korea; 3Central Advanced Research and Engineering Institute, Hyundai Motor Company, 37 Cheoldobangmulgwan-ro, Uiwang-si 16082, Korea; kimhjoo@gmail.com

**Keywords:** seat comfort, automobile seat, self-shape adjustable seat, pneumatic system, driver’s experience

## Abstract

The seat comfort of automobiles is one of the significant factors for determining the driver’s fatigue, emotional experience, and individual space (which captures their individuality, rather than just a means of transportation in modern society). Conventional automobile seats could not provide seating comfort suitable for all drivers, in the form of seats that fit each driver’s body type and the difficulty of meeting individual needs. This study proposes self-shape adjustable (the SSA seats) seats that improve the sitting comfort, safety, and secure the stability, by adjusting shape fit to the driver’s body type. The SSA seats transforms the seat itself, in a way that improves the distribution of contact pressure and reduces sitting fatigue, with the pneumatic system. The transformed seats provide better sitting comfort and safety than the conventional automobile seat, by providing a seat shape suitable for the body shape of all users. It was verified that the SSA seats, proposed in this paper, have a uniform and improved pressure distribution, compared to the conventional seat, in various sitting postures; the contact area between the seat and user is enlarged, and the pressure concentrated on the ischial bone is lowered. In addition, it was proven (through user evaluation) that quantitative evaluation verification was the same as qualitative evaluation results.

## 1. Introduction

Vehicle seat comfort, along with other major performance factors, is a key factor influencing a purchase. In terms of driving performance, noise, vibration, and collision safety, the technology gap between major manufacturers has recently narrowed. In addition, automobile technologies, such as autonomous driving, electric vehicles, and connectivity, have provided more convenience to people, allowing them to focus on subjective quality. As a result, major manufacturers are working hard to improve convenience, such as seat comfort, to differentiate their products. This means that vehicle comfort, convenience, and safety are more important than vehicle performance and efficiency.

All advanced technologies have been integrated into vehicles for comfort and safety. In this regard, since safety and comfort are related to seat design and functionality, many researchers are paying more attention to car seats closely related to passenger safety and comfort. Many studies have been performed that research comfort, seat design, and sitting environments [1,2]. Unlike when cars were just transportation, sitting comfort has developed into an important problem. There are many evaluation studies on sitting comfort, mainly related to the relationship between comfort, sitting posture, and other related factors. De Looze et al. (2003) [3] studied the relationship between comfort and discomfort through objective measurement and subjective evaluation, as did Kyoung and Nussbaum (2008) [4,5] and Carcone and Keir (2007) [6]. Reed et al. (1994) [7] studied the effect of driving posture, while Helander et al. (1987) [8] and Sanders and McCormick (1987) [9] studied the effect of pressure distribution on contact areas. Colich et al. (2004) [10] estimated the comfort of automobile seats, by more accurately measuring the contact area and pressure of the seats. As many studies have shown, vehicle comfort and discomfort are closely related to seat contact area and pressure. They act as the main parameters that determine seating comfort. Many studies have also studied parameters to improve the comfort of the automobile seats, and David et al. (1995) [11] is one of the improved studies and has been researched to evaluate the effectiveness of intelligent seating systems with air cell seats. In the study, the effect of pressure distribution and contact area on sitting comfort was also evaluated with a comfort index. Some researchers, Grieko (1986) [12], studied sitting postures to estimate physical and medical effects, especially Rajput and Abboud (2007) [13], who studied improper effects via foot postures.

Unfortunately, although these studies have found and studied the effects between driving posture and comfort and health, they have not presented an improved design or method for developing the comfort of the seat, depending on the seating environment. In this study, to improve the comfort of passenger seats, self-adjustable seats (SSA seats) are proposed as a new model for automobile seat. SSA seats provide passengers with an optimized shape and pressure to improve the comfort of the seats, according to their body shape, posture, and weight. The area, shape, and contact force between the passenger and the seat are determined by the passenger’s posture and weight. In other words, the variables that affect wear and comfort depend on the passenger’s body type and weight. Therefore, the parameters of comfort may be defined as body shape, sitting posture, and weight. To evaluate the seating comfort of SSA seats, the average pressure, pressure distribution, maximum pressure, and contact area of SSA seats and conventional car seats were compared and used. As Kazushige (2001) [14] mentioned, these parameters are the main factors influencing seating comfort.

One of the main factors, the feeling of the floor reflects the impact stiffness when seated, and the feeling of foam hardness reflects the rigidity of the driving environment where you sit for a long time. This value is estimated by measuring the foam stiffness, sciatic pressure, and contact area between the passenger and seat. The SSA seats system consists of an air pack seat system and a pneumatic system. Cells in the air pack system provide seats of appropriate shape to the passenger’s hips and distribute seat pressure across the entire contact area. The pneumatic system regulates the internal air pressure of the air packaging cell, so that it can function as the air packaging system itself. The prediction of SSA seats functionality and seat comfort was tested and subject-rated. These results will help develop new applications and features in car seats and related industries.

## 2. Self-Shape Adjustable Seat System

The SSA seats consist of two independent units. This device consists of polyvinyl packaging and polyethylene foam interior materials. Each device is independently connected to the pneumatic system and can be moved individually. The internal materials constituting each unit were determined through tests for candidates of various materials. The unit configuration and material candidates are shown in Figure 1, and the initial concept of the system is shown in Figure 2. In the concept system, the seating part is composed of coffee grains for a soft sitting feeling, as the grains can give the passenger more comfort and expect a stable sitting feeling. In Figure 3, the next development version consisting of a grain type with a size of 2–3 mm expanded the polystyrene, but all grain types were excluded because they did not meet the repeatability and durability criteria. So, finally, sponge material was selected. The sponge is polyethylene foam and is a material having similar material properties, and the existing automobile sheet is made of polyethylene foam. Therefore, in this paper, polyethylene foam (S-50), with a density of 50 kg/m${}^{3}$ in the configuration of the seats, was selected.

The SSA seats are divided into a control and driving device. A drive device that operates in contact with a user has already been mentioned above, and Figure 4 shows the control device that controls it; Figure 5 shows the pneumatic circuit block diagram. The control flowchart is shown in Figure 6. The pneumatic circuit block diagram represents the control flow of the SSA sheet, and the control board is displayed on the control board. The main role of the pneumatic system is to suck air inside the air packaging cell and blow air into shape when restoring shape. The OVEM ejector is controlled by the MCU; the air inlet and damage are caused by the compressor. The solenoid valve is an air gate between the compressor and packaging seat system and is controlled by the MCU. When the pneumatic system is activated, the suction of the air inside the air packaging cell is started through the OVEM ejector, and the internal air pressure can be maintained when the internal air pressure of the pneumatic cell reaches 0.4 bar. The OVEM ejector itself can measure the air pressure of the air pack and is controlled by the MCU, by measuring the set point of 0.4 bar value.

The development of the SSA system began with a manual system and developed into an active system. In the initial state, the air volume inside the air pack was determined using a switch; in the active system, the internal air pressure was measured using the OVEM ejector, and the internal air volume was controlled at the measurement set value. The system has a flow rate of up to 100 L/min and uses a rated 6280 rpm BLDC motor. The reason for using a 100 L/min flow compressor in this study is that it requires a high-capacity compressor to test variable materials (including granular materials) and various types of air packs. Upon further work, the SSA system will be optimized and developed to suit the vehicle environment, as well as equipped with an appropriate compressor. In the case of passive systems, air suction was possible; when the air pack shape was restored, air injection was impossible, making it difficult to ensure consistency in the restoration shape. Because the injection was possible, it was possible to secure the consistency of the fast restoration time and form.

## 3. Experiment

### 3.1. Environment and Participants

Seven healthy students participated in the experiment. All participants were male and had no musculoskeletal disorders. Their average age, height, weight, and driving experience were 24.5 years old (1.8 years old), 172.5 cm (5.7 years old), 65.8 kg (12.4 years old), and 5.3 (2.5 years old) years, respectively. The numbers in parentheses mean standard deviations. They were carefully selected to represent young and healthy male students because the first consumer to buy a car was a man of this age and had no prejudice against seat comfort [15]. The SSA seats system mounted on the testbed is shown in Figure 7 and Figure 8, respectively.

A prototype of the front seat pan cushion of a large sedan, mass-produced in Korea, was made with a driving device for a testbed. The prototype was also applied with changes to install the drive device control system in the existing sedan seats. In this study, only static sitting tests of various sitting postures were included. This is because, before performing the dynamic test, the power source and the SSA seats system must be optimized to meet the dynamic test conditions of the vehicle. In the current stage of the SSA seats system, vibration and dynamic environmental tests were not performed while driving because the use of power and compressors was not suitable for vehicle test conditions. It will be processed and carried out during additional work. A fully mounted testbed is shown in Figure 9.

### 3.2. Material Selection Test

For SSA sheet development, sheet convenience, durability, and shape resilience were compared with existing car seats. Seat comfort was evaluated mainly based on whether the distribution of contact pressure between the seat and user was locally concentrated. Although repeatability can be tested in this study, durability was excluded in this study because durability cannot be tested through several test cases and repetitions. In addition, the seat type restoration test visually determined whether the seat type was restored to its original shape 10 min after completion of the seat.

All candidate materials were evaluated through testing. Therefore, the smart sheet material was determined through uniform ranking evaluation of pressure distribution among materials with no clear problem in resilience. As shown in Figure 10, sponge materials were selected among the five other materials. Thereafter, the seating convenience was compared for four different densities of the selected sponge, as shown in Figure 11.

The sponge consists of polyethylene. As shown in Figure 10, it was superior to any other candidate, in terms of pressure distribution capacity, load reduction, and contact area expansion. Therefore, as shown in Figure 11, only sponge materials were retested to select the best density for the SSA sheet.

### 3.3. Elasticity Measurement Test

As shown in Figure 12, a system was constructed to measure the elasticity of the sheet foam. Figure 12 represents the entire elasticity measurement system, and Figure 13 represents the system configuration of the measurement system. Existing studies have shown the density and elasticity of the foam, in which the components of the car seat determine the comfort that passengers can feel. In addition, the relationship between density, elasticity, and comfort can provide optimized comfort to passengers through quantitative evaluation. The results of this test were used as evaluation data for the material selection test conducted in Section 3.2.

### 3.4. Data Collection Procedures and Processing

To compare the SSA seats with regular car seats, the pressure distribution between the backrest and pan was measured via a pressure mat (X3 PX20048*48, XSENSOR Co, Canada), shown in Figure 14. The pressure distribution ratio to the pan in a regular car seat is 74:19 (1995) [16], and the load on the pan is almost three times higher than the rear. In this test, the pressure distribution was measured for about 40–60 s, from the start of pneumatic suction of the SSA seats until the pneumatic and shape changes reached a stable state. Two-dimensional images and numerical data of the contact area and pressure between the user and seat were measured with a pressure mat, and the air pressure inside the air pack was measured through an OVEM ejector. In the case of the final data used for the analysis, the average of the values (measured twice) was used, in consideration of measurement and filter error.

### 3.5. Subjective Ratings

In this paper, numerical values are used as an index for evaluating the comfort of seats. This is because existing studies rely on qualitative analysis to evaluate only comfort. Therefore, they required a lot of time, resources, and participants in the car seat development process. Only numerical values are used to reduce these efforts, but subjective grades are needed first to confirm numerical results. The questionnaire consists of 15 items, as seen in Table 1. Items are grouped into four specific configurations: stress, shape, stiffness, and comfort.

Pressure refers to the pressure felt in the seat, when sitting in the SSA seats. Shape support means the amount of coverage for the shape of the hips and thighs. Firmness means the stiff feeling of the seat. Comfort means the overall feeling in each part of the body. Each item evaluates the feeling that can be distinguished from the SSA seats. The items of the survey were reconstructed by referring to other studies, shown in Table 2. In most studies, the pan consists of four parts (left and right of the hip and thigh, pressure, stiffness, and comfort), which were estimated through the questionnaire. They focused their efforts on determining the index that evaluates comfort because it was difficult to quantitatively evaluate comfort. Therefore, in this study, instead of the index evaluated by passengers, we tried to determine the standard for evaluating comfort by measuring changes in the contact area, peak pressure, and average pressure. Therefore, this questionnaire is an auxiliary method for verifying standards.

The questionnaire was implemented in a total of four test cases, as shown in Figure 15. Case 1 is the posture of the passenger, leaning against the back seat. Case 2 is the posture of the passenger not leaning against the back seat. In Cases 3 and 4, the driver’s posture for the conditions of Cases 1 and 2, respectively.

### 3.6. Data Analysis

The focus of the data analysis is how much the surface contact area between the user and the seat is expanded, and the surface peak contact pressure is reduced through the SSA seats. As described above, the contact area and maximum pressure are indicators for evaluating the comfort of the seat [21]. Therefore, the numerical data of pressure and contact areas have been mainly treated as 2D image data.

## 4. Results

Subjective usability evaluation was performed on three, finally selected, models, as shown in Figure 16, among the proposed models. Model A has a 50 kg/m${}^{3}$ polyethylene foam, with a side pedestal capable of further supporting the thigh (auxiliary). Model B used a wider version, with the plane of Model A (without side bolsters). Model B shows the advantage of covering passengers with heavy bodies or passengers sitting with wide thighs open. Model C, based on material test candidate sponge + granule, has a polypropylene granule (PP short time) ball layer inside the air packaging sheet on only the thigh part. In other words, the ball layer does not exist in the buttocks area, as shown in Figure 17. This PP layer provides stability through rigidity stronger than polyethylene (PE short time) foam. In addition, subjective evaluation of existing car seats was additionally performed, and it was used as control group to evaluate three candidate seat models.

Figure 17 shows the experimental results of each seat model participant, and Table 3 shows the numerical data of Figure 18. AVG refers to the average pressure of the seat pan portion of the pressure sensor, CPP refers to the cushion peak pressure, CCA refers to the cushion contact area, and LOAD refers to the total load of the measured pressure sensor (Kolich, 2004 [18]. In addition, init, 8 kg, and critical (on the top row) refer to the initial state, measured by 8kg each for the expected driving state and pressure, when the pressure reaches its peak. The seat contact load measured while driving is about 110% of the static state. Therefore, when the user’s load in the static state is 80 kg, the seat comfort in the dynamic state may be evaluated by adding a weight of 8 kg to measure the seat pressure. Delta is the result of subtracting critical from init, so this value helps to easily show the strain.

Model A is superior to other models in CCA (order of increasing contact part), and Model B and Model C are superior to Model A in CPP (lowest pressure order). Average quantitative differences for LOAD are described in Table 4. It can be seen that a total of four postures are evaluated in Table 4. The four postures are the leaning passenger, erect passenger, the leaning driver, and the erect driver postures. The above four postures were classified to evaluate both the driving situation and seat position of the non-driver, while sitting in the driver’s seat. In addition, the posture of the non-driver’s seat may represent the posture of all users, when using the autonomous vehicle. Figure 18, Table 3, Figure 19, and Table 4 are about Case 1 for all participants. Therefore, the former was used to select the optimal model, and the latter was used for detailed analysis. However, all cases were tested for all models. According to Figure 18 and Table 3, Models A and C perform better than Model B. For Models A and C, the contact area with CCP is improved by 34%, 110%, 31%, and 110%, respectively (improved means pressure reduction and increase area). The hip part of the SSA seats is about 20% larger than the thigh part. The division of the hip and thigh is shown in Figure 19. This is because there is a backbone in the hip joint, and the bone has the highest pressure when measuring pressure. When dividing the measured values from the data measured, as shown in Figure 19, into three parts, the top 1/3 is regarded as the hip part, and the bottom 2/3 is regarded as the thigh part. This is because the seat pan pressure distribution consisted of 73.84% of the hips, 18.48% of the thighs, and the rest consisted of pan pedestals, and the ratio was 7.67%. (Westlander et al. 1995 [16]).

The hips are treated as more important than anything else, in terms of CCP, because pressure concentrated on the hips regenerates, compared to the thighs. As mentioned earlier, this is because the ischial bone of the hip joint area shows the highest contact pressure. As shown in Figure 18, the pressure around the ischial bone was evenly distributed at the threshold for all users of all models, so that the concentrated pressure part disappeared. Compared to the existing sheet, shown on the far left, it can be seen that the locally-concentrated pressure, from Models A to C, is relatively low. According to these results, it can be confirmed that the SSA seats provide improved seat comfort, by dispersing the pressure caused by the weight of the user applied to the seat pan. Figure 18 shows the same results in Figure 18. The peak pressure delta, in Cases 2 and 4, is a posture characteristic. As we already know, the contact pressure around the thigh is more focused on the erect posture.

The SSA seat’s contact area distribution and expansion mean that it provides passengers with more comfort than conventional seats. To confirm this, a questionnaire has been implemented, as shown in Table 1. The questionnaire is based on Table 2. The participants answered that they felt comfortable and stable when the SSA seats were in the critical state, especially the C model gave them a more comfortable feeling. Model C has a PP layer on the thigh and provides more stability to the occupant, due to rigidity. The correlation between the experimental results and comfort is confirmed by the participant’s rating, and, in conclusion, the SSA seats system provides better comfort to passengers than the conventional automobile seat.

## 5. Conclusions

The SSA seats system, proposed in this study, has the effect of expanding the contact area and lowering the concentrated pressure, by changing the shape of the seat according to the body shape of the occupant. This provided better comfort and seating stability to the occupant, compared to the existing seat and was proven through user evaluation and quantitative test evaluation. In particular, it was confirmed through quantitative test evaluation that the pressure concentration near the ischial bone, which was seen in the existing car seat, was greatly improved. As mentioned earlier, it can be confirmed that the proposed system can provide riding comfort for any posture of both drivers and non-drivers, by performing the evaluation for both drivers and non-driver, through testing according to various sitting postures.

The proposed SSA seats system provides improved comfort by changing the physical properties of the seat form and various shape design methods, as well as by controlling the contact pressure between the seat and each part of the body; it responds to dynamic postures, as well static postures. It is also possible to improve on a specific part of the body. In addition, the possibility of developing a system for the safety of users in specific driving situations, such as vehicle turning or sudden stops and starts, was also confirmed. Through future research, not only the currently developed seat pan part but also the seat back and various rest parts can be developed as a system that secures users’ safety and comfort, by expanding the system. In addition, it will be possible to shorten the reaction time and improve noise by composing four air packs and improving the pneumatic actuator and system expansion; thus, improving the internal structure of the air pack could respond to individual elasticity by quantile also. In the control part, it is also possible to strengthen shape-response functions, such as critical point control and memory function.

In this paper, we proposed three models of the SSA seats, and model C is, finally, selected via experiment and subjective ratings. However, those models have stated prototypes, so that the level of comfort and stability would be improved by further work and development with researches. Propagation to the back of the seat and additional functions, such as correcting seated postures or emergency actions, would be a supplementary part of the further work. The propagation could be also head, arm, and leg rest. The approach in this paper, for improving sitting comfort, is to make the seat comfort evaluation easier than before, by showing the consistency between the quantitative and qualitative evaluations and replacing the qualitative evaluation with a quantitative evaluation. In quantitative evaluation, errors inevitably follow, and discussions to resolve them must continue. However, in this paper, since the average value of the results of three measurements is used for quantitative evaluation for the aforementioned approach, errors in the quantitative evaluation are not dealt with in this paper.

## Figures and Tables

**Figure 1 sensors-21-07959-f001:**
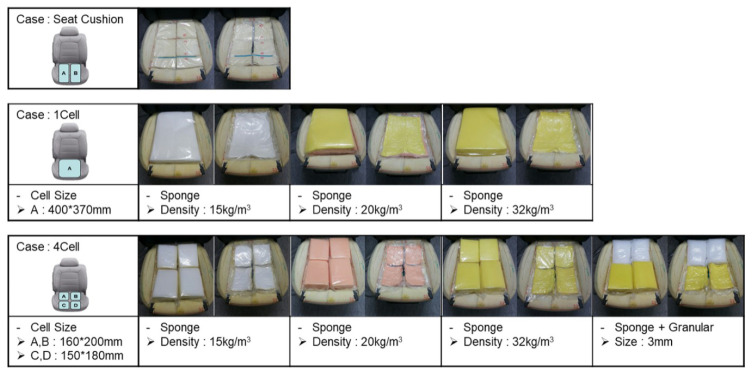
Material selection and unit configuration test.

**Figure 2 sensors-21-07959-f002:**
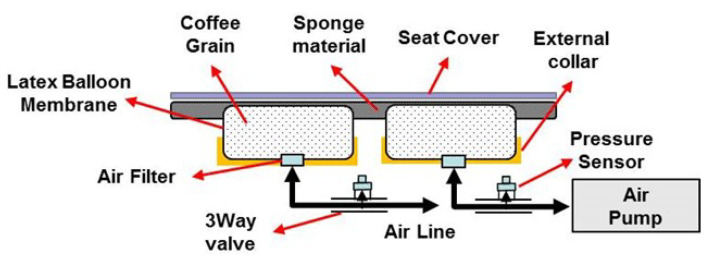
Initial concept of SSA seats system.

**Figure 3 sensors-21-07959-f003:**
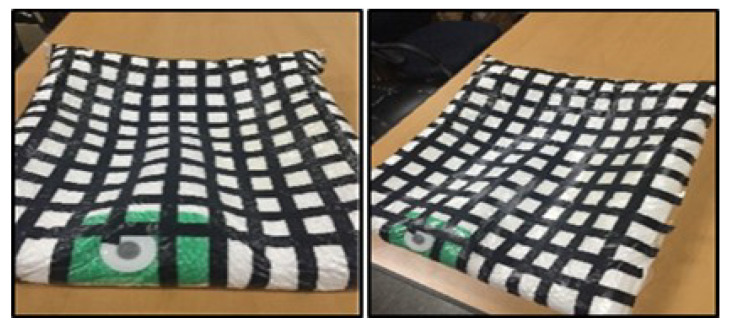
The Proto type of the SSA seats with a grain type. Front (**left**), Diagonal (**right**).

**Figure 4 sensors-21-07959-f004:**
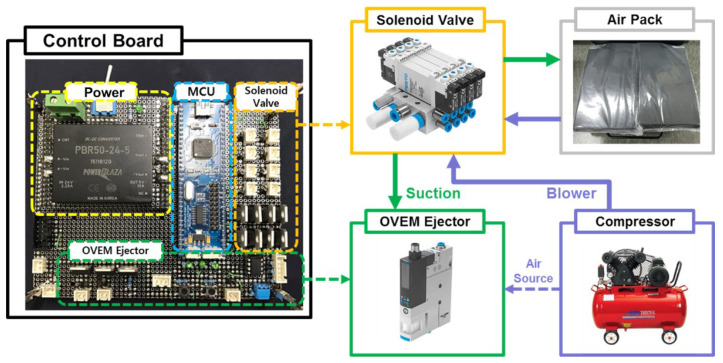
Configuration of the pneumatic system.

**Figure 5 sensors-21-07959-f005:**
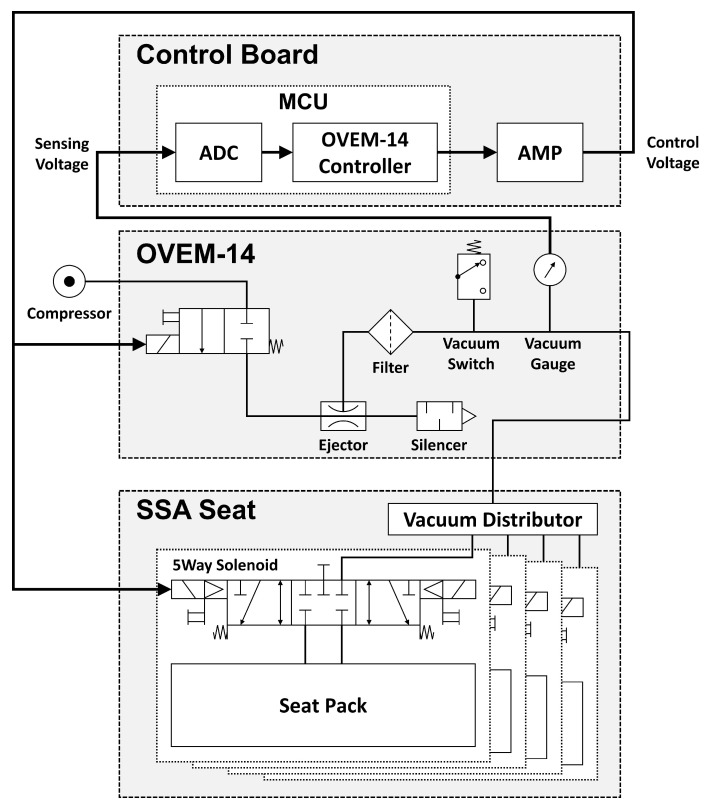
Pneumatic circuit block diagram of SSA system.

**Figure 6 sensors-21-07959-f006:**
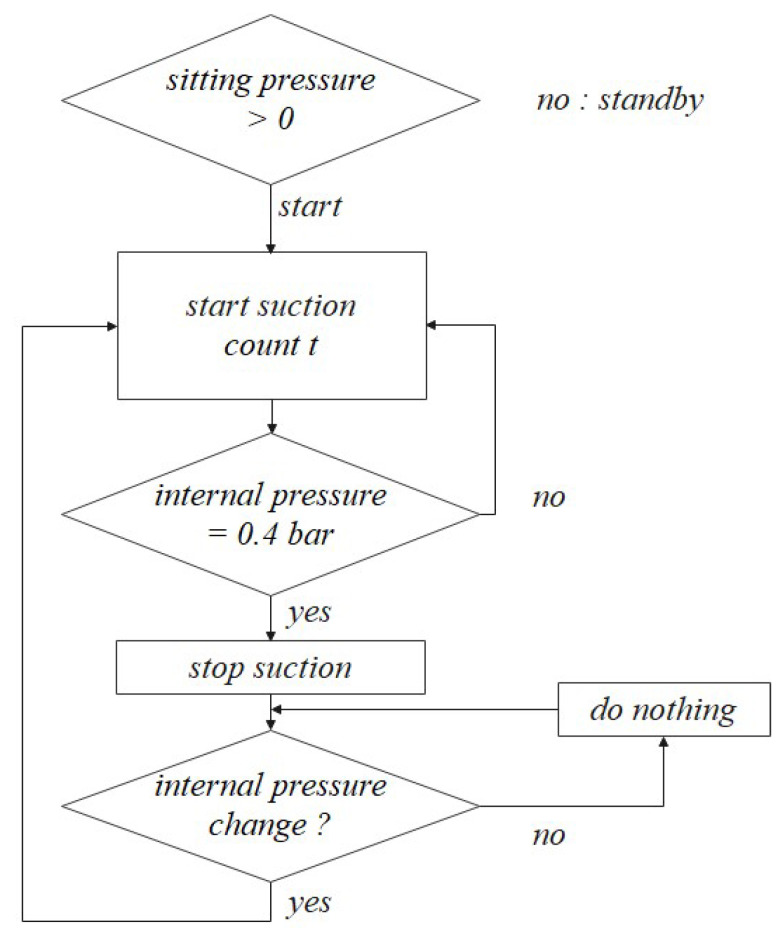
Actuation process chart of SSA system.

**Figure 7 sensors-21-07959-f007:**
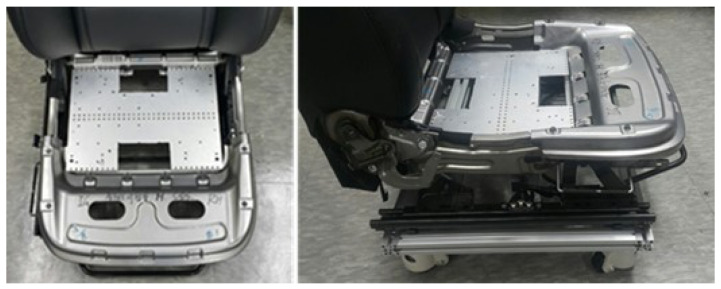
Testbed of SSA seats system. Front (**left**), Side (**right**).

**Figure 8 sensors-21-07959-f008:**
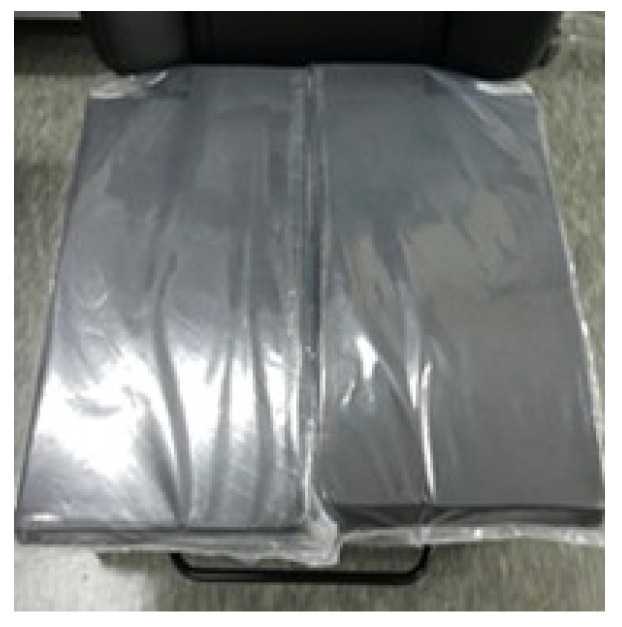
SSA seats equipped on test bed.

**Figure 9 sensors-21-07959-f009:**
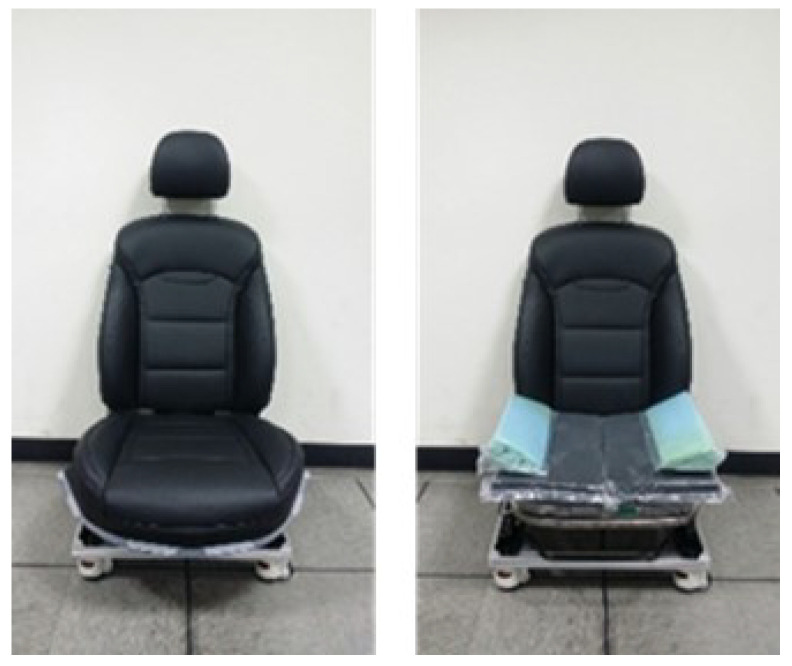
Air-packed, cell-integrated type of SSA seats system, with (**left**) and without (**right**) seat covers.

**Figure 10 sensors-21-07959-f010:**
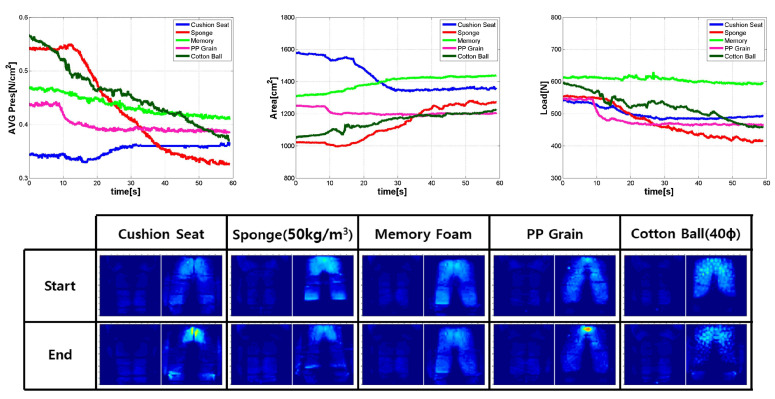
The result of the material selection test (all materials). Avg. pressure (**Up-Left**), Contact Area (**Up-Center**), Load (**Up-Right**), Init & Critical pressure map of each material seat.

**Figure 11 sensors-21-07959-f011:**
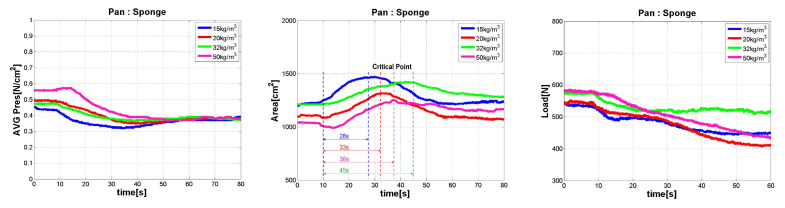
The result of material selection test Avg. pressure (**Left**), Contact Area (**Center**), Load (**Right**), (only sponge with various densities).

**Figure 12 sensors-21-07959-f012:**
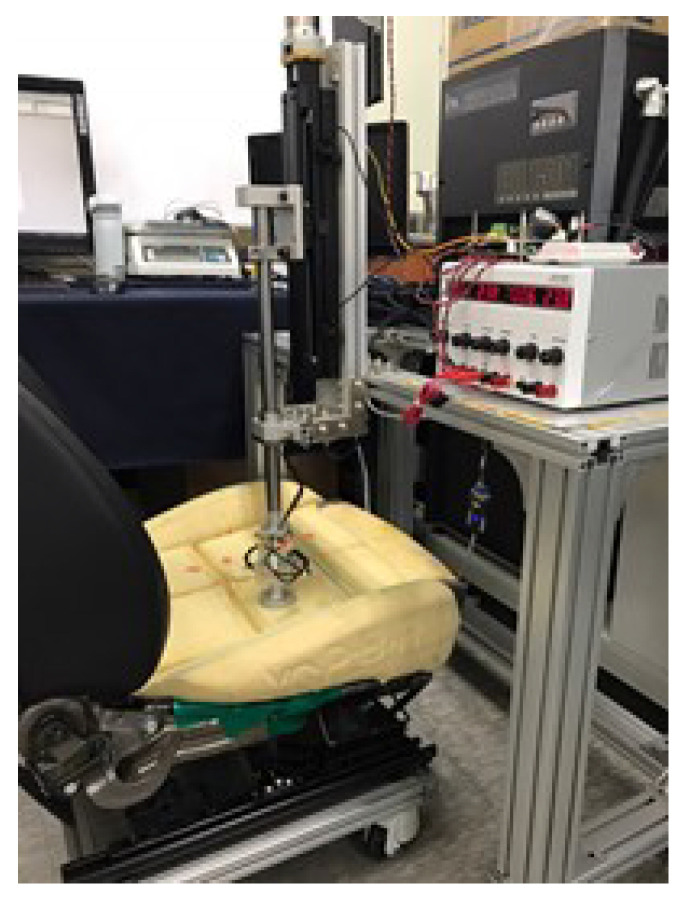
Elasticity measuring system.

**Figure 13 sensors-21-07959-f013:**
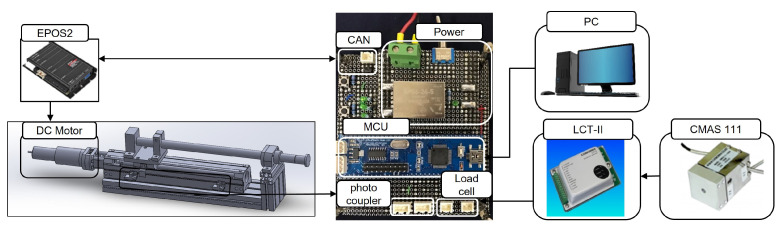
Configuration of elasticity measuring system.

**Figure 14 sensors-21-07959-f014:**
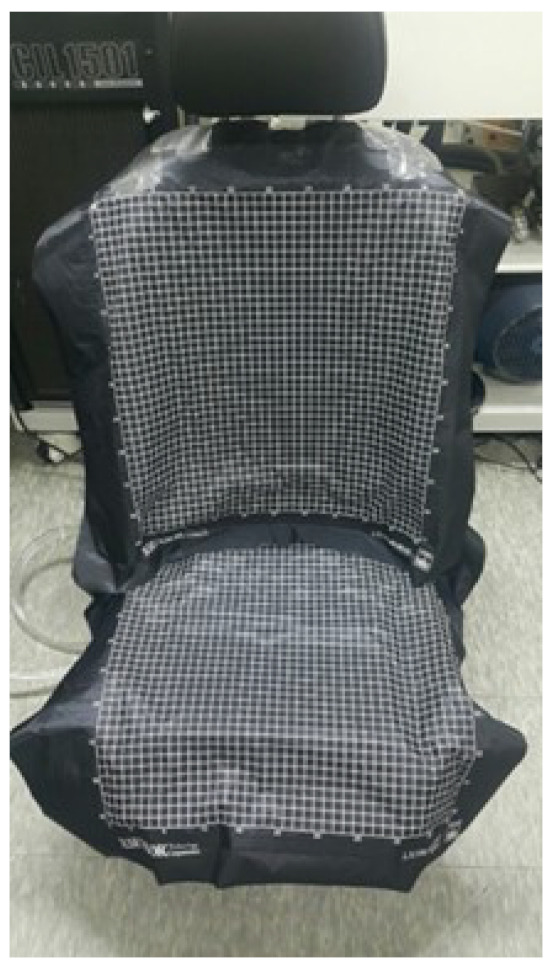
Pressure mat setting with SSA seats.

**Figure 15 sensors-21-07959-f015:**
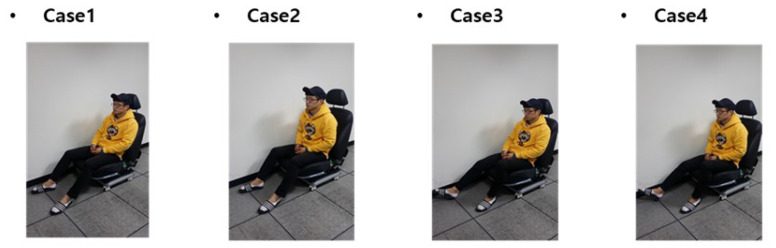
Test case, according to sitting posture.

**Figure 16 sensors-21-07959-f016:**
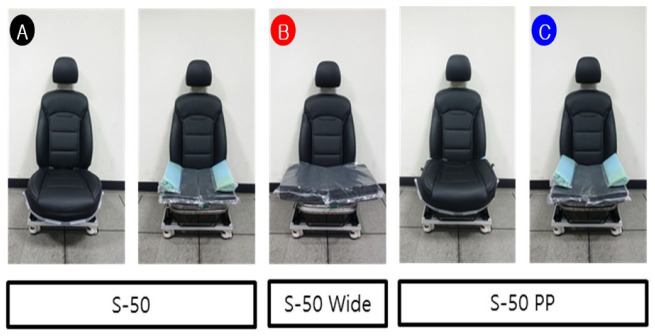
3 Final models (A: S-50, B: S-50 Wide, C: S-50 PP).

**Figure 17 sensors-21-07959-f017:**
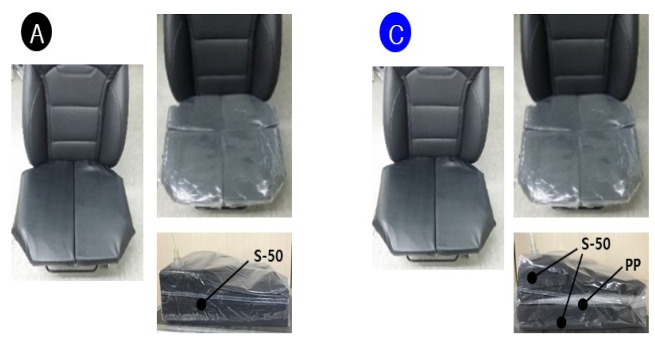
Configuration of pan layer of SSA seats (only for Model A and C).

**Figure 18 sensors-21-07959-f018:**
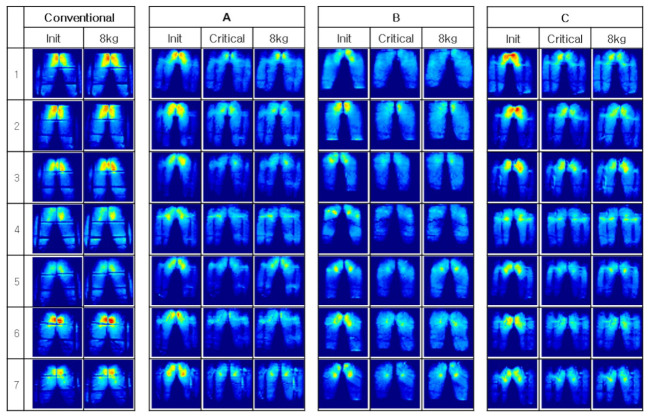
Measured data: change of pressure distribution.

**Figure 19 sensors-21-07959-f019:**
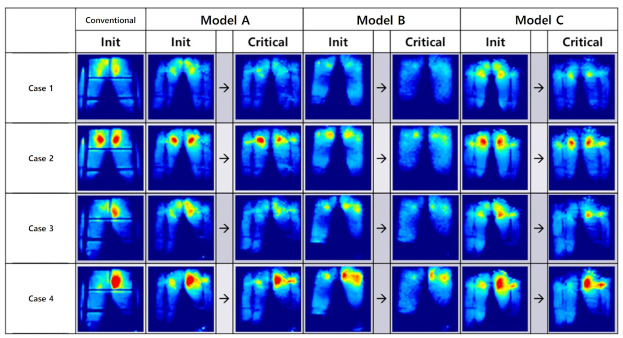
Measured data: change of pressure distribution for each sitting posture.

**Table 1 sensors-21-07959-t001:** The questionnaire for subjective ratings.

Items		Objective		
		Conventional	Model A	Model C
Pressure	Pressure (stress) on thigh area			
	Pressure (stress) on lateral area			
	Pressure (stress) on ischial/buttocks area			
	Pressure (stress) on tailbone			
Shape Support	Amount of thigh support			
	Amount of cushion lateral support			
	Amount of ischial/buttocks support			
Firmness	Thigh cushion firmness			
	Lateral cushion firmness			
	Pan cushion firmness			
Comfort	Thigh comfort			
	Cushion lateral comfort			
	Cushion tailbone comfort			
	Ischial/buttocks comfort			
	Overall seat comfort			

**Table 2 sensors-21-07959-t002:** Reference list of questionnaire.

	D. Ng et al. (1995) [11]		Kolich (2000) [17]		Kolich and White (2004) [18]		Smith el al (2006) [19]		Baba et al (2008) [20]	
Seat Pan				Cushion Tailbone	Pressure					
	Buttocks	Support			Ischial buttocks	Amount of support				
					Thigh	Amount of support				
					Cushion Lateral	Amount of support				
	Seat Cushion	Firmness			Cushion	Firmness	Cushion	Firmness		
			Cushion Tailbone	Comfort	Cushion Tailbone	Comfort				
			Ischial	Comfort	Ischial	Comfort			Buttock	Comfort
			Thigh	Comfort	Thigh	Comfort			Thigh	Comfort
			Cushion Lateral	Comfort	Cushion Lateral	Comfort				
					Overall Seat Cushion	Comfort				
Seat Back					Back Tailbone	Pressure	Lumbar	Pressure		
					Back lateral	Amount of support				
					Upper-back	Amount of support			Upper-back	Support
	Lumbar	Support	Lumbar	Amount of support	Lumbar	Amount of support	Lumbar	Support	Lumbar	Support
	Seat back	Firmness			Seat back	Firmness				
							Lumbar	Stiffness		
							Backrest	Firmness		
			Lumbar	Comfort	Lumbar	Comfort				
			Upper-back	Comfort	Upper-back	Comfort				
			Back Lateral	Comfort	Back Lateral	Comfort				
			Back Tailbone	Comfort	Back Tailbone	Comfort				
					Overall Seatback	Comfort				
Overall					Overall Seat	Support				
					Overall Seat	Comfort	Overall	Discomfort	Overall	Discomfort

**Table 3 sensors-21-07959-t003:** Measured data: pressure, contact area, and load, when in init, critical, and long driving situations.

Model		Conventional		Model A				Model B				Model C			
**Participant**	**Item**	**Init**	**8 kg**	**Init (I)**	**Critical (C)**	**Delta (C-I)**	**8 kg**	**Init (I)**	**Critical (C)**	**Delta (C-I)**	**8 kg**	**Init (I)**	**Critical (C)**	**Delta (C-I)**	**8 kg**
1	AVG	0.39	0.41	0.38	0.33	−0.05	0.37	0.40	0.36	−0.04	0.40	0.40	0.36	−0.04	0.40
	CPP	1.39	1.47	1.54	1.05	−0.49	1.20	1.61	1.04	−0.57	1.05	1.61	1.02	−0.59	1.04
	CCA	1439	1561	1475	1606	131	1669	1545	1630	85	1654	1545	1622	77	1656
	LOAD	582	640	561	530	−31	618	618	584	−34	662	618	587	−31	662
2	AVG	0.38	0.41	0.38	0.34	−0.04	0.36	0.45	0.40	−0.05	0.43	0.38	0.33	−0.05	0.38
	CPP	1.44	1.52	1.26	1.03	−0.23	1.02	1.39	1.00	−0.39	0.96	1.62	1.03	−0.59	1.05
	CCA	1395	1467	1475	1614	139	1703	1298	1398	100	1458	1414	1520	106	1616
	LOAD	530	601	561	549	−12	613	537	502	−35	614	584	559	−25	627
3	AVG	0.37	0.41	0.34	0.27	−0.07	0.31	0.40	0.36	−0.04	0.39	0.36	0.34	−0.02	0.38
	CPP	1.42	1.57	1.14	0.88	−0.26	0.94	1.04	0.81	−0.23	0.95	1.44	1.02	−0.42	1.18
	CCA	1259	1377	1290	1372	82	1509	1106	1172	66	1243	1296	1372	76	1479
	LOAD	466	565	439	370	−69	468	467	466	−1	562	442	422	−20	485
4	AVG	0.40	0.44	0.40	0.33	−0.07	0.38	0.41	0.31	−0.10	0.37	0.42	0.35	−0.07	0.41
	CPP	1.15	1.17	1.09	0.97	−0.12	1.02	1.24	0.74	−0.51	0.79	1.08	0.92	−0.16	1.00
	CCA	1348	1425	1358	1448	90	1527	1109	1170	61	1253	1327	1430	103	1501
	LOAD	539	627	543	478	−65	580	557	501	−56	615	455	363	−92	464
5	AVG	0.32	0.35	0.33	0.29	−0.04	0.36	0.35	0.31	−0.04	0.34	0.37	0.33	−0.04	0.41
	CPP	0.98	1.03	1.10	0.83	−0.27	1.02	1.16	0.90	−0.26	1.16	1.19	0.85	−0.34	1.08
	CCA	1304	1356	1274	1304	30	1379	1243	1293	50	1416	1148	1212	64	1240
	LOAD	420	480	417	375	−42	495	438	502	−37	479	420	398	−22	510
6	AVG	0.40	0.43	0.38	0.31	−0.07	0.34	0.39	0.33	−0.06	0.37	0.42	0.39	−0.03	0.43
	CPP	1.71	1.78	1.57	1.07	−0.50	1.09	1.31	0.90	−0.41	0.95	1.22	1.06	−0.16	1.09
	CCA	1362	1393	1350	1475	125	1582	1329	1437	108	1524	1240	1312	72	1359
	LOAD	543	594	517	450	−67	537	523	477	−46	564	526	512	−14	584
7	AVG	0.38	0.41	0.41	0.31	−0.10	0.35	0.36	0.31	−0.05	0.36	0.37	0.33	−0.04	0.40
	CPP	1.49	1.62	1.40	1.06	−0.34	1.08	1.12	0.98	−0.14	1.05	1.23	0.97	−0.26	1.12
	CCA	1217	1285	1246	1404	158	1517	1220	1329	109	1400	1156	1212	56	1243
	LOAD	466	531	512	442	−70	530	445	416	−29	498	431	400	−31	502
Average	AVG	0.38	0.41	0.37	0.31	−0.06	0.35	0.39	0.34	−0.05	0.38	0.39	0.35	−0.04	0.40
	CPP	1.37	1.45	1.30	0.98	−0.32	1.05	1.27	0.91	−0.36	0.99	1.34	0.98	−0.36	1.08
	CCA	1340	1409	1353	1460	108	1555	1264	1347	83	1421	1304	1383	79	1442
	LOAD	507	577	507	456	−51	549	512	478	−34	571	497	463	−34	548

**Table 4 sensors-21-07959-t004:** Reference list for the questionnaire.

Model		Conventional	Model A		Model B		Model C	
**Participant**	**Item**	**Init**	**Init (I)**	**Critical (C)**	**Init (I)**	**Critical (C)**	**Init (I)**	**Critical (C)**
CASE 1	AVG	0.45	0.38	0.33	0.40	0.34	0.44	0.34
	CPP	1.27	1.13	1.03	1.20	0.82	1.26	1.06
	CCA	1295	1325	1462	1114	1153	1317	1420
	LOAD	584	497	478	443	391	579	485
CASE 2	AVG	0.52	0.45	0.44	0.46	0.43	0.50	0.44
	CPP	1.96	2.05	2.07	1.42	1.06	1.91	1.63
	CCA	1316	1350	1461	1154	1227	1379	1512
	LOAD	683	605	640	530	528	695	664
CASE 3	AVG	0.43	0.40	0.36	0.42	0.38	0.45	0.37
	CPP	1.49	1.38	1.13	1.29	0.97	1.51	1.39
	CCA	1183	1201	1333	1020	1088	1282	1429
	LOAD	510	478	476	432	414	575	531
CASAE 4	AVG	0.52	0.48	0.44	0.51	0.44	0.53	0.46
	CPP	2.99	2.85	2.24	1.73	1.46	2.24	2.10
	CCA	1216	1227	1412	1100	1156	1329	1441
	LOAD	635	587	621	559	503	701	668

## Data Availability

Not applicable.

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
