# Peer review of "A Development of the Self Shape Adjustment Cushion Mechanism for Improving Sitting Comfort"

_sensors, 2021, doi:10.3390/s21237959_

Round 1
Reviewer 1 Report
This paper proposes a smart seat that improves the seating comfort, safety and secures the stability by adjusting shape fit to the body. The main idea may be of interest to the scientific community. The paper is well organized, and the methodology is duly described. The experiments were properly described, and the results are according to the methodology. The manuscript needs minor improvements to be published in this journal. Some comments are listed below:
- As you may know, the recommended length of an Article is more than 16
journal pages. If you still want to keep it in the form of Article, please extend the manuscript or Communication is another option. The following link is the author instruction page of Sensors, which might be helpful:
https://www.mdpi.com/journal/sensors/instructions
- The technical description of the control unit should be increased, since it is a fundamental part of the intelligence of the system.
- You said “Smart seats transform the seat itself in a way that improves the distribution of contact pressure and reduce sitting fatigue by internal air control system”. But, It is not clear if a control algorithm is implemented by the MCU, such as a PID, to perform the air control adjustment.
- It is necessary to include a mathematical model of the dynamics of the process or define the system's transfer function.
- I think the authors didn't show enough evidence to prove the good working ability of the system. So, I suggest a systematic investigation on the signal processing before publish.
- An intense study about the performance of the system together with a thorough discussion about pros and cons when compared with other approaches could result in a good paper to publish.
- The discussion of errors should be provided, and the advantages that can be obtained from the proposed approach, with measurements that support the discussion.
Reviewer 2 Report
This paper is about an automobile seat cushion. The authors investigated and analyzed the factors related to the comfort of the passengers.
- In section 2, the author proposed the initial concept with three figures. Some of them are excluded as mentioned “but the all the grain types were excluded due to could not meet the criterion of repeatability and 88 durability.” Additionally, the authors added design parameters in section 3. t is confusing what the model and the smart seat are and how the smart seat works. Clarify what the final design is and define all the input and output factors before experiments.
- The “smart” in the smart seat seems to mean that the shape of the seat changes according to a passenger. However, in the paper, the authors look for a comfortable seat with a fixed shape. Then what does the smart mean in this paper, exactly?
- Check the standard deviation of the driving experience. Statically, 3.5 SD with 6 positive values from 6 samples and 2 of average value is too larger.
- Additional explanations are required for Figures 8 and 9. For example, the physical meaning and effects on comfortableness of the avg. area. and load in figure 8 and figure 9. Enlarge the captions. It is too small to read.
- Check the manuscript on page 10. It looks that this is not the final version. For example,
-“According to Figure 16 and Table 3, model A and model C shows the best performance than model C”
- The performance of models A and C was analyzed and compared as shown in Figure 17 and Table 4 / however the figure and table show the results of model A, B, and C.
- And to verify the pressure distribution and expansion of the contact area so that the smart seat provides comfort to passengers, a questionnaire was implemented as shown in Table 1. The questionnaire is developed based on Table 2 Participants answered that the smart seat felt comfortable and stable when it was in the final Steady state, and that the C model in particular gave a more comfortable feeling.”
- In the proposed system, the proposed seat needs an air compressor which is large and heavy for a passenger car. Is it reasonable to use an air compressor for seats in a passenger car?
- typo : [?] in line 110
Round 2
Reviewer 1 Report
This paper proposes a smart seat that improves the seating comfort, safety and secures the stability by adjusting shape fit to the body. The main idea may be of interest to the scientific community. The paper is well organized, and the methodology is duly described. The experiments were properly described, and the results are according to the methodology. Thank you for taking my feedback. The authors have worked hard to improve the manuscript; it is much clearer than the old version. I recommend this paper to be published in this journal. Here I would like to list some minor comments are listed below to help further improve the manuscript quality.
- There is one visible syntax error in the abstract: line 13
- Carry out an exhaustive review of English grammar to the entire manuscript.
Reviewer 2 Report
The revised paper needs professional English editing. There are too many typos and errors in grammar. Sentences should start with a capital letter. Do not start sentences with "And" or "But"
For example, "So, the formet set" in 242, "sponges is consist" in 153, "And the sponges are developed to polyethylene foam because the polyethylene is similar to cushion of the conventional seat. And it has 50kg/m3 density. So, it called S-50 in this paper. " in 89 and so on.
